# *MBTPS2* mutations cause defective regulated intramembrane proteolysis in X-linked osteogenesis imperfecta

Uschi Lindert[1,*], Wayne A. Cabral[2,*], Surasawadee Ausavarat[3,4,*,†], Siraprapa Tongkobpetch[3,4], Katja Ludin[5], Aileen M. Barnes[2], Patra Yeetong[3,4,†], Maryann Weis[6], Birgit Krabichler[7], Chalurmpon Srichomthong[3,4], Elena N. Makareeva[8], Andreas R. Janecke[7,9], Sergey Leikin[8], Benno Röthlisberger[5], Marianne Rohrbach[1], Ingo Kennerknecht[10], David R. Eyre[6], Kanya Suphapeetiporn[3,4,*], Cecilia Giunta[1,*], Joan C. Marini[2,*] & Vorasuk Shotelersuk[3,4]

Osteogenesis imperfecta (OI) is a collagen-related bone dysplasia. We identified an X-linked recessive form of OI caused by defects in *MBTPS2*, which encodes site-2 metalloprotease (S2P). *MBTPS2* missense mutations in two independent kindreds with moderate/severe OI cause substitutions at highly conserved S2P residues. Mutant S2P has normal stability, but impaired functioning in regulated intramembrane proteolysis (RIP) of OASIS, ATF6 and SREBP transcription factors, consistent with decreased proband secretion of type I collagen. Further, hydroxylation of the collagen lysine residue (K87) critical for crosslinking is reduced in proband bone tissue, consistent with decreased lysyl hydroxylase 1 in proband osteoblasts. Reduced collagen crosslinks presumptively undermine bone strength. Also, proband osteoblasts have broadly defective differentiation. These mutations provide evidence that RIP plays a fundamental role in normal bone development.

[1] Division of Metabolism, Connective Tissue Unit and Children's Research Center, University Children's Hospital Zurich, Zurich 8032, Switzerland. [2] Section on Heritable Disorders of Bone and Extracellular Matrix, National Institute of Child Health and Human Development, National Institutes of Health, Bethesda, Maryland 20892, USA. [3] Center of Excellence for Medical Genetics, Department of Pediatrics, Faculty of Medicine, Chulalongkorn University, Bangkok 10330, Thailand. [4] Excellence Center for Medical Genetics, King Chulalongkorn Memorial Hospital, The Thai Red Cross Society, Bangkok 10330, Thailand. [5] Center for Laboratory Medicine, Department of Medical Genetics, Kantonsspital Aarau, Aarau 5001, Switzerland. [6] Department of Orthopedics and Sports Medicine, University of Washington, Seattle, Washington 98195, USA. [7] Division of Human Genetics, Medical University of Innsbruck, Innsbruck 6020, Austria. [8] Section on Physical Biochemistry, National Institute of Child Health and Human Development, National Institutes of Health, Bethesda, Maryland 20892, USA. [9] Department of Pediatrics I, Medical University of Innsbruck, Innsbruck 6020, Austria. [10] Institute of Human Genetics, Westfälische Wilhelms University, Münster 48149, Germany. * These authors contributed equally to this work. † Present addresses: Division of Nuclear Medicine, Department of Radiology, Faculty of Medicine Siriraj Hospital, Mahidol University, Bangkok 10700, Thailand (S.A.); Division of Human Genetics, Department of Botany, Faculty of Science, Chulalongkorn University, Bangkok 10330, Thailand (P.Y.). Correspondence and requests for materials should be addressed to J.C.M. (email: oidoc@helix.nih.gov).

Osteogenesis imperfecta (OI), or 'brittle bone disease', is a well-known heritable skeletal dysplasia. Affected individuals have low bone mass and increased fracture susceptibility. OI is genetically heterogeneous. Most cases have autosomal-dominant mutations in the type I collagen genes, COL1A1 or COL1A2 (ref. 1). More recently, multiple genes causing mostly autosomal recessive forms of OI were delineated, generating a new paradigm for OI as a collagen-related disorder. The products of these genes (CRTAP, LEPRE1, PPIB, FKBP10, PLOD2, SERPINH1, BMP1, SERPINF1, IFITM5 and SPARC) interact with collagen post-translationally for prolyl 3-hydroxylation, folding, processing, fibrillogenesis, cross-linking and/or mineralization[1,2]. No cases of X-linked OI have been reported.

Regulated intramembrane proteolysis (RIP) involves cleavage of membrane-spanning regulatory proteins by proteases within the plane of the membrane[3]. The three intramembrane cleaving protease (iCLiP) families include serine proteases, site-2 metalloprotease (S2P) and aspartyl proteases. The best-described RIP-mediated signalling in humans involves sequential cleavage of diverse substrates by site-1 protease (S1P), a serine protease encoded by membrane-bound transcription factor peptidase, site 1 (MBTPS1), and S2P, encoded by membrane-bound transcription factor peptidase, site 2 (MBTPS2). S1P and S2P are located in the Golgi membrane. They cleave regulatory proteins transported from the ER membrane in times of ER stress or decreased sterol metabolites, releasing mature N-terminal fragments that shuttle to the nucleus and activate gene transcription. These substrates include old astrocyte specifically induced substance (OASIS)/cAMP responsive element binding protein 3-like 1 (CREB3L1), a transcription factor expressed in astrocytes and osteoblasts, activating transcription factor 6 (ATF6), a component of the unfolded protein response pathway and sterol regulatory element binding protein (SREBP), involved in cholesterol synthesis[4–6]. MBTPS2 missense mutations were previously identified in the related dermatological conditions IFAP/BRESHECK (ichthyosis follicularis, atrichia, and photophobia, OMIM #308205)[7,8] and keratosis follicularis spinulosa decalvans (KFSD, X-linked, OMIM #308800)[9].

In this report, we present two independent OI pedigrees with an X-linked inheritance pattern, and without symptoms of IFAP/KFSD. Linkage analysis and next-generation sequencing (NGS) identified a novel MBTPS2 missense mutation in each pedigree. The resulting substitutions are in or near the S2P NPDG motif crucial for metal ion coordination[10]. Mutant S2P protein is stable but cleavage or activation of S2P substrates OASIS and ATF6, respectively, is impaired, consistent with reduced proband collagen secretion. Osteoblasts with mutant S2P have defective differentiation, and bone tissue collagen from one proband has decreased hydroxylation of the critical crosslinking lysine residue. These X-linked OI-causing mutations extend RIP functioning to normal bone formation.

## Results

### Clinical features.
Family I. From an extended Thai OI pedigree, 6 of 12 affected males were examined (Fig. 1a,b and Supplementary Table 1). The proband (P1/I; V7) had prenatal fractures of ribs and long bones (Fig. 2a). At 2 years, his physical exam was notable for moderate short stature, blue sclerae, pectus carinatum and bowing of lower extremity long bones. His L2–L4 bone mineral density z-score = − 4.7 (0.315 g cm$^{-2}$). Affected adult males had fractures beginning during gestation, short stature, white sclerae, variable scoliosis and pectal deformity, striking tibial anterior angulation and generalized osteopenia (see Case Reports).

Family II. A non-consanguineous German pedigree has two affected males, the 26-year-old proband (P1/II) and his 68-year-old maternal uncle (P2/II) (Fig. 2c; Supplementary Fig. 2a and Supplementary Table 1). As a newborn, the proband was noted to have bowing of humeri, radii and tibiae, and fractures of femora, ribs and clavicles, as well as generalized osteopenia (Fig. 2b and Supplementary Fig. 2b). Both patients had numerous fractures of upper and lower extremity long bones during childhood, but fractures declined post puberty. Both have white sclerae, kyphoscoliosis with anterior vertebral wedging and short stature, with final height of 5–6-year-old males. The uncle has severe pectus excavatum.

None of the affected individuals had ichthyosis follicularis, alopecia, photophobia, intellectual disability or seizures.

### Mutation identification.
Sequencing of a set of genes involved in OI and bone fragility (ALPL, BMP1, COL1A1, COL1A2, CREB3L1, CRTAP, FKBP10, IFITM5, LEPRE1, LRP5, PLOD2, PLS3, PPIB, SERPINF1, SERPINH1, SP7, TMEM38B and WNT1) did not reveal functional changes in the probands of either pedigree. Each proband had a neutral COL1 single-nucleotide polymorphism (SNP) that was inherited from his healthy father (Supplementary Fig. 3). Interestingly, the urinary lysyl pyridinoline (LP) to hydroxylysyl pyridinoline (HP) ratio, which has been suggested as a diagnostic marker for OI[11], was markedly elevated in an adult from Family I and two adults from Family II (0.305–0.450 patients versus controls 0.211 ± 0.008) and in the index case from Family I (0.319 versus paediatric controls 0.208 ± 0.030) (Supplementary Table 2). The LP/HP ratio was not elevated in a carrier from each OI Family, or in a case of IFAP. The elevated LP/HP ratio is similar to findings in a mouse model of type IX OI[12], although X-OI probands have normal PPIB sequences.

X-chromosome inactivation studies of five obligate carriers from Family I showed a skewed pattern of inactivation (Supplementary Fig. 4). Linkage analysis of the X-chromosome in Family I defined a 21.6-Mb critical region on Xp22 from markers DXS7108 to DXS1067 (LOD score 3.31) (Fig. 1b and Supplementary Table 3). Comparative genomic hybridization (CGH) analysis did not reveal imbalances in the critical region, which was then sequenced by NGS in affected male IV17 and unaffected male IV10. After excluding known SNPs, and variants present in the unaffected relative, two candidates remained. A mutation in MBTPS2 exon 11, c.1376A>G (nucleotide: NM_015884.3, GRCh38) predicting a p.N459S substitution (protein: NP_056968.1), was confirmed by Sanger sequencing of proband DNA (Fig. 2c), segregated with OI in the pedigree (Fig. 1b) and was absent in 644 X chromosomes of unrelated Thai controls (181 females, 282 males). A FAM48B1 variant, c.532G>A (p.V178M), occurred in 4 of 22 alleles from Thai controls (Supplementary Table 4).

In Family II, SNP-genome-wide linkage analysis supported X-linked recessive inheritance, despite normal X-inactivation (0.70) in the mother of P1/II, with a positive LOD score in a 30-Mb region of chromosome X between rs11094708 and rs5906168 (Supplementary Fig. 1a and Supplementary Table 5). In view of the modest positive LOD score, the entire X-exome was sequenced by NGS, resulting in two variants with population frequencies below 0.5% within the 30-Mb critical region. The first variant, a hemizygous c.1778G>A (p.R593H) missense SNP in CCDC120, was detected in proband P1/II but not in his affected uncle (P2/II). The second variant, a hemizygous c.1515G>C mutation in MBTPS2 (p.L505F; Supplementary Fig. 1b), co-segregated with the phenotype and was also found in a heterozygous state in the proband's mother (Fig. 2d).

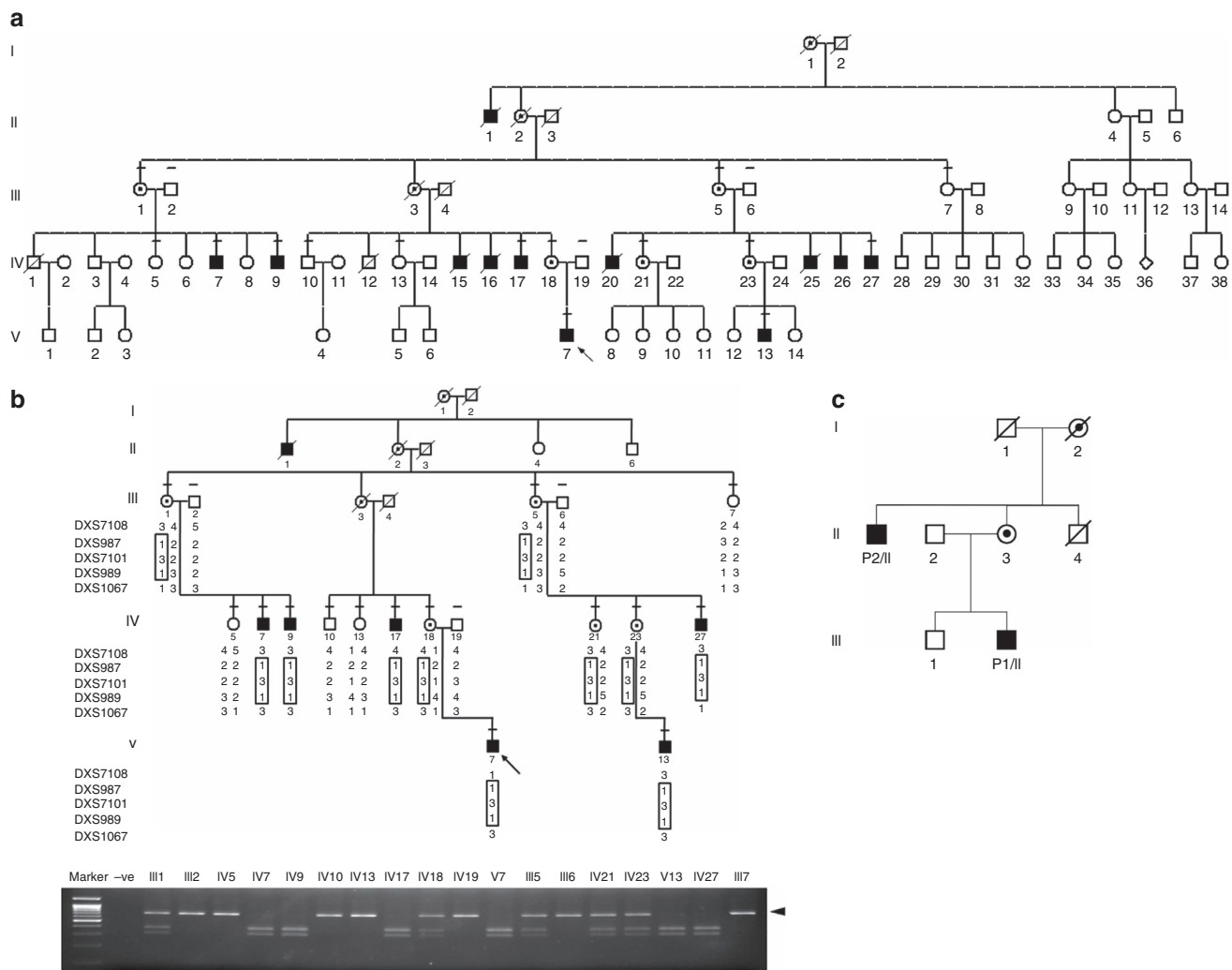

**Figure 1 | Extended pedigree of Family I with linkage and segregation analyses.** (**a**) Extended pedigree of Family I. The pedigree contains 12 affected members (blackened symbols), with seven living and five deceased (diagonally crossed). Clinical assessments and blood samples were obtained from 18 individuals (short horizontal bars above symbols), including six affected. Dotted circles denote obligate carriers. Arrow indicates Proband P1/I. (**b**) Linkage and segregation analysis. Thai pedigree shows linkage of mutation with X chromosome and cosegregation of the mutation with phenotype. X-chromosomal markers are shown from Xpter to Xcen (full list of markers in Supplementary Table 3). Linked markers in squares are shown in relation to the members' status. The *MBTPS2* c.1376A>G (p.N459S) mutation segregated with phenotype in all affected members. The mutation introduces a *Bsg*I cleavage site, resulting in 375 and 305 bp bands. The normal allele generates the undigested 680 bp band. Marker = 100 bp marker; −ve, negative control with no DNA added; an arrowhead indicates the 680 bp band. (**c**) Pedigree of Family IIA. The segregation analysis was done on all family members still alive at the time of sampling, that is all but individuals I/1, I/2 and II/4. The index patient P1/II and his maternal uncle P2/II show very similar clinical findings compatible with the diagnosis osteogenesis imperfecta type III/IV. The mother of P1/II is of normal height, whereas the maternal grandmother is only 155 cm (<3rd centile), but does not show any signs of osteogenesis imperfecta.

Multiple programs (Polyphen2, SIFT, Provean, Mutation-Taster) predict both *MBTPS2* (S2P) variants p.N459S and p.L505F as pathogenic since they are likely to interfere with function (Supplementary Table 6). Residues N459 and L505 are highly conserved in S2P, as identified by ClustalX (Fig. 2e), as well as ClustalW and Boxshade (for p.L505F). Neither of the *MBTPS2* mutations, c.1376A>G (p.N459S) and c.1515G>C (p.L505F) are present in the Leiden Open Variation Database (LOVD), Exome Aggregation Consortium (ExAC), the Exome Variant Server (EVS), 1000 Genomes Browser and dbSNP Build 144 variant databases (Supplementary Table 6). N459 is the first residue of the NPDG motif, which is important for metal ion coordination[10].

**Effect of *MBTPS2* mutations on transcripts and protein.** *MBTPS2* transcript levels were not reduced in fibroblasts with OI,

IFAP (p.R429H) or KFSD (p.N508S) mutations, by real-time RT–PCR (Fig. 3a). In addition, S2P protein displayed normal stability in these cells on immunoblot (Fig. 3b).

**Effect of mutant S2P on RIP functions.** We examined the impact of S2P substitutions on activation of RIP substrates OASIS, ATF6 and SREBP (Fig. 4a–c). On western blot, mature intracellular OASIS cleavage fragment (50 kDa, S1P/S2P cleaved) was decreased, while the 55-kDa cleavage product (S1P cleaved) was increased, in fibroblasts with each mutation and in S2P p.L505F osteoblasts, indicating S2P cleavage impairment in OI probands comparable to IFAP and KFSD cells (Fig. 4a). Treatment with ALLN (N-acetyl-leucyl-leucyl-norleucinal) to prevent degradation of OASIS cleavage products increased the total OASIS detected in all fibroblast and osteoblast cell lines, as well as the relative proportion of fully cleaved OASIS in normal

control, IFAP and KFSD fibroblasts and normal control osteo-blasts, more than in OI cells (Fig. 4a). Treatment with tunicamycin to induce ER stress had no further effect on OASIS cleavage.

An ATF6 reporter was co-transfected with *MBTPS2* expression constructs containing normal, OI-causing (p.N459S or p.L505F) or IFAP/BRESHECK (p.R429H)[7,8] sequences into CHO-M19

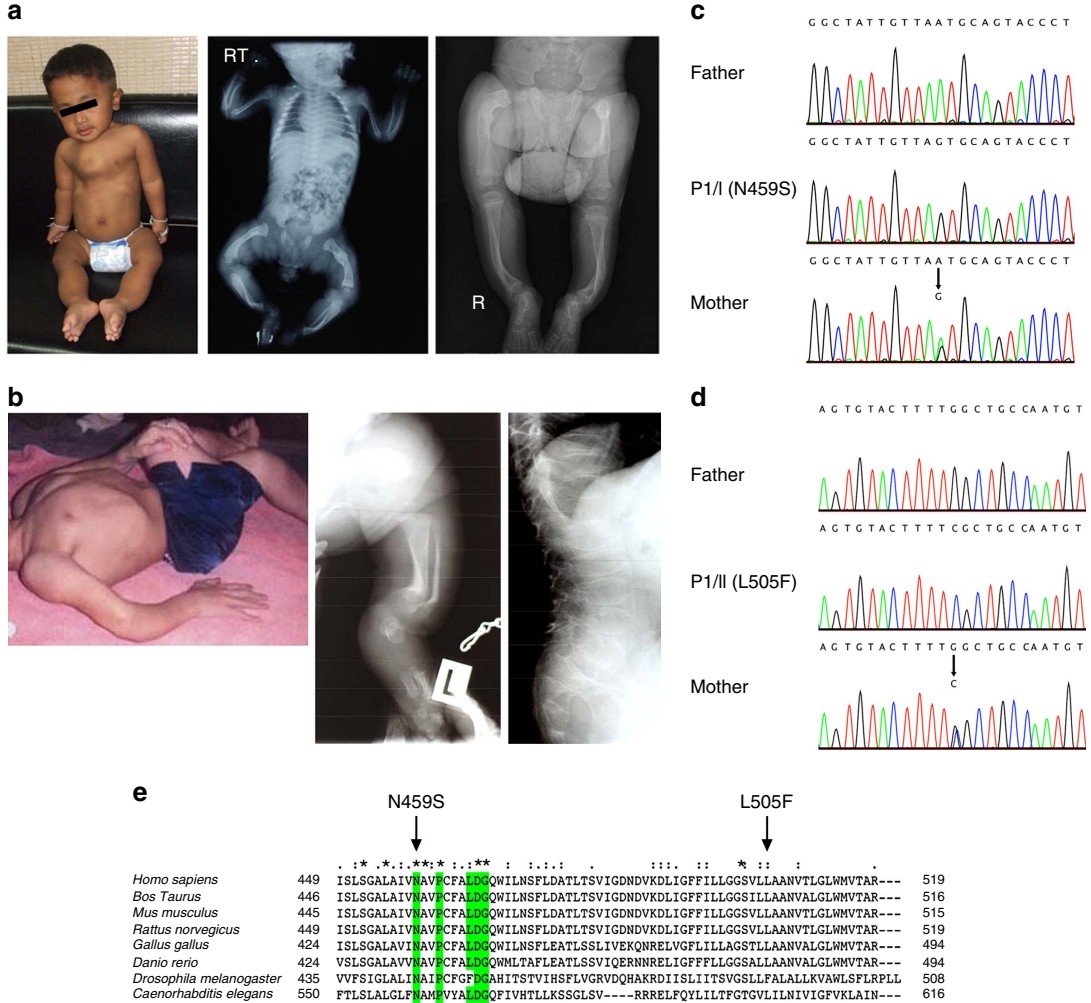

**Figure 2 | Clinical features of probands and mutation identification.** (**a**) Left, Proband 1/I had blue sclerae, *pectus carinatum* and anteriorly bowed legs at 2 years of age. Centre, radiographs showed fractures of ribs, humeri, femora and tibiae on day 1 of life. Right, undertubulation and minimal cortex of lower long bones at 2 years of age. (**b**) Left, Proband 1/II at age 20 years presenting with significant rhizomelia of upper and lower extremities. Centre, radiographs at birth showed bowed tibiae and fibulae (left leg shown). Right, radiographs revealed kyphoscoliosis with anterior vertebral wedging, and flat, biconcave vertebral bodies with significant osteoporosis at 13 years of age. (**c**) *MBTPS2* genomic DNA sequence reveals the c.1376A > G mutation (p.N459S) in proband P1/I and his heterozygous mother, but not in the proband's father. (**d**) Sequence of P1/II genomic DNA shows the *MBTPS2* c.1515G > C mutation (p.L505F), which also occurs in heterozygous form in his mother, but is not present in the proband's father. (**e**) Species comparison of the S2P amino-acid sequence according to ClustalX. Residues shaded in green denote the NPDG motif required for enzymatic active site metal ion coordination.

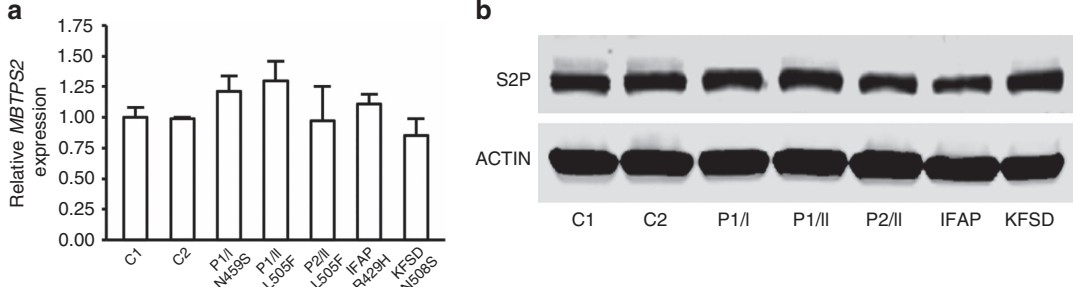

**Figure 3 | Expression of *MBTPS2* in OI and IFAP fibroblasts.** (**a**) Quantification of *MBTPS2* transcripts by real-time RT–PCR shows no significant differences. (**b**) S2P protein levels are normal in OI, IFAP and KFSD fibroblasts, as detected by western blot. No significant differences in *MBTPS2* expression were detected between control and proband fibroblasts. Error bars, s.d.

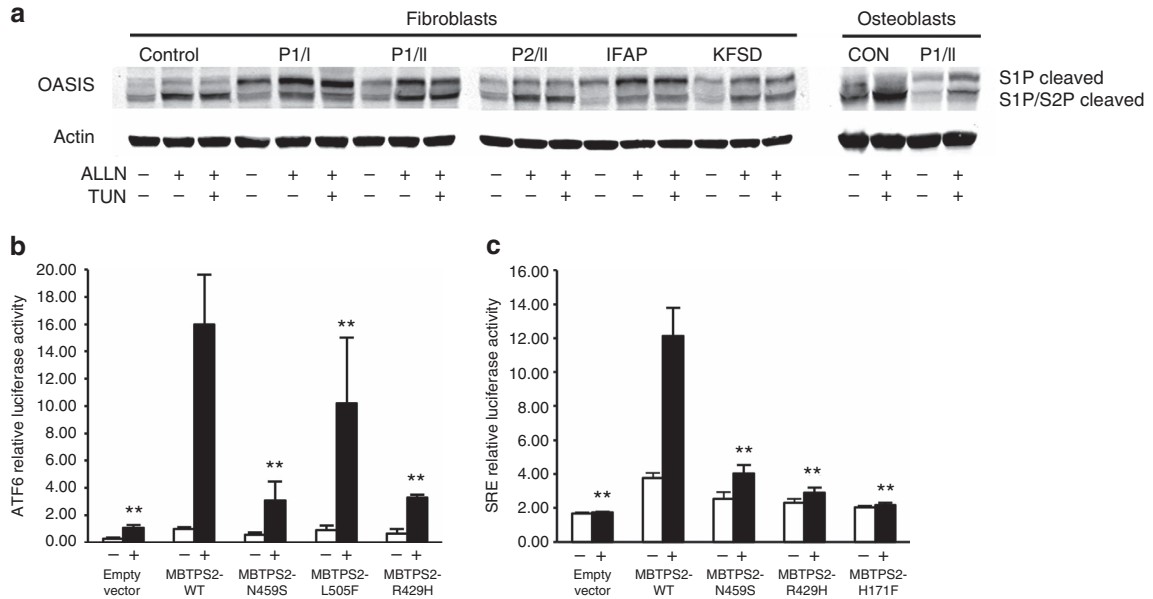

**Figure 4 | Functional studies of the *MBTPS2* variants. (a)** Western blots of OASIS cleavage by S1P and S2P. Control fibroblast and osteoblast lysates show cleavage of OASIS to its mature active form (S1P/S2P cleaved). Cells from OI, IFAP and KFSD probands contain increased partially cleaved OASIS (S1P cleaved) due to impaired cleavage by S2P (TUN, tunicamycin; ALLN, *N*-[*N*-(*N*-acetyl-L-leucyl)-L-leucyl]-L-norleucine). **(b)** Luciferase assays of *MBTPS2* activation of ATF6 reporters upon induction of stress ( + ). Renilla luciferase was used to normalize the transfection efficiency. WT, N459S, R429H, and L505F indicate that cells were transfected with normal or mutant *MBTPS2* constructs. \*\**P*, wild-type versus empty vector, N459S, R429H, and L505F < 0.001 by ANOVA. **(c)** Luciferase assays of *MBTPS2* activation of SRE reporters on induction of stress by depletion of sterols ( + ). Renilla luciferase was used to normalize the transfection efficiency. The results were represented as mean ± s.d. WT, N459S, R429H and H171F indicate that cells were transfected with normal or mutant *MBTPS2* constructs. \*\**P*, wild-type versus empty vector, N459S, R429H and H171F < 0.001 by ANOVA. Error bars, s.d.

cells lacking *MBTPS2* (Fig. 4b). The transcriptional activation of *ATF6* after tunicamycin-induced ER stress was significantly decreased in CHO-M19 cells expressing mutant constructs (p.N459S, p.L505F, p.R429H) versus normal *MBTPS2* ($P < 0.01$) (Fig. 4b and Supplementary Table 7), as measured by luciferase activity. Interestingly, the depression of *ATF6* activation from p.L505F transfection was less marked than with p.N459S or IFAP-causing p.R429H. This is similar to the degree of impaired OASIS cleavage in untreated mutant fibroblasts, in which p.N459S and IFAP cells had relatively more uncleaved OASIS than did p.L505F cells, although all mutants did significantly impair cleavage versus control.

Studies of SRE reporter co-transfection with *MBTPS2* expression constructs were done with only p.N459S and IFAP/KFSD mutants (Fig. 4c, Supplementary Table 8). Luciferase activation was suppressed in all cells expressing mutant constructs in sterol-free media.

**Effect of defective S2P on type I collagen biochemistry.** Since OI is a collagen-related bone dysplasia and fully processed OASIS is reported to activate *Col1a1* transcription[13], we examined type I collagen biochemistry in fibroblasts and osteoblasts with mutant S2P. Collagen transcripts were variable in mutant cells, with decreased *COL1A1* transcripts detected in fibroblasts of OI proband 1/I (p.N459S), IFAP (p.R429H) and KFSD (p.N508S) fibroblasts, but not in OI fibroblasts or osteoblasts with the S2P p.L505F mutation (Fig. 5a).

Fibroblasts from OI probands secreted a significantly reduced amount of type I collagen (20–73% of control), as did IFAP and KFSD fibroblasts ($\approx$40%) (Fig. 5b). Proband osteoblasts were insufficient for this assay. Electrophoretic migration of proband collagen (Fig. 5c) was normal, consistent with normal thermal stability ($T_{\mathrm{m}}$) and helical resistance to proteolytic digestion at physiologic temperature (Supplementary Fig. 5a,b). Collagen

deposited into matrix in culture by S2P p.N459S fibroblasts from P1/I had a decreased proportion of collagen containing mature cross-links (10% versus 22% of collagen deposited by normal control fibroblasts) (Supplementary Fig. 5c), suggesting that hydroxylation of the type I collagen K87 residue crucial to crosslinking may be impaired.

**Effect of mutant S2P on bone tissue and cells.** Tissue and osteoblast characteristics of OI were demonstrated in a bone sample from P1/II (p.L505F). Osteoblasts differentiated in culture revealed impaired expression of transcripts related to osteoblast maturation and RIP pathways (Fig. 6). Alkaline phosphatase (*ALPL*) expression is about half of control in mid- to late differentiation. OASIS (*CREB3L1*) expression was significantly reduced throughout osteoblast differentiation. Mature OASIS normally complexes with SMAD4 to upregulate expression of matrix genes[14]. However, *SMAD4* transcripts were reduced throughout p.L505F osteoblast differentiation, underlying the reduced expression of the proteoglycan matrilin-1 (ref. 15). Interestingly, *COL1A1* expression in P1/II osteoblasts was comparable to control (Fig. 5a).

Collagen extracted directly from P1/II bone tissue had less than half the normal level of hydroxylation of the lysine (K87) critical for collagen crosslinking in both collagen alpha chains (43% versus 90% control in α1(I), and 7% versus 43% control in α2(I)) (Fig. 7a and Supplementary Fig. 6). This is consistent with both the proband's increased urinary LP/HP ratio and reduced crosslinked collagen deposited into matrix. However, hydroxylation of collagen carboxyl-terminal crosslink residues K930/933 was normal in each chain (Fig. 7a and Supplementary Fig. 6). The enzyme responsible for modification of the crosslinking lysine (K87), lysyl hydroxylase 1 (LH1, encoded by *PLOD1*), was substantially reduced in S2P p.L505F osteoblast lysates, although the level of CyPB/PPIB, a foldase critical to LH1 function, was

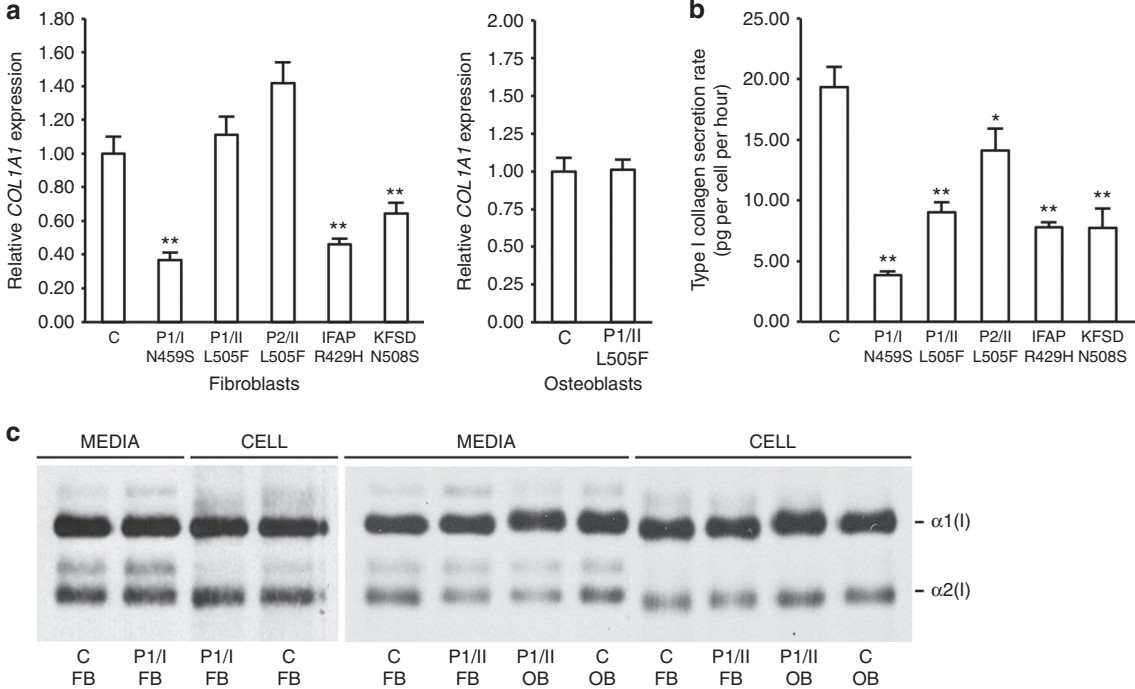

**Figure 5 | Effects of *MBTPS2* mutations on type I collagen. (a)** Expression of *COL1A1* in normal control (C), OI proband, IFAP and KFSD fibroblasts, and OI and control (C) osteoblasts. **(b)** Total procollagen secretion by fibroblast lines harbouring *MBTPS2* mutations is decreased compared with normal control (C) fibroblasts. **(c)** Steady-state type I collagen analysis. The electrophoretic migration of collagen alpha chains synthesized by OI proband fibroblasts (FB) and osteoblasts (OB) was equivalent to normal control (C) fibroblast collagen. *$P < 0.05$, **$P < 0.001$ by *t*-test. Error bars, s.d.

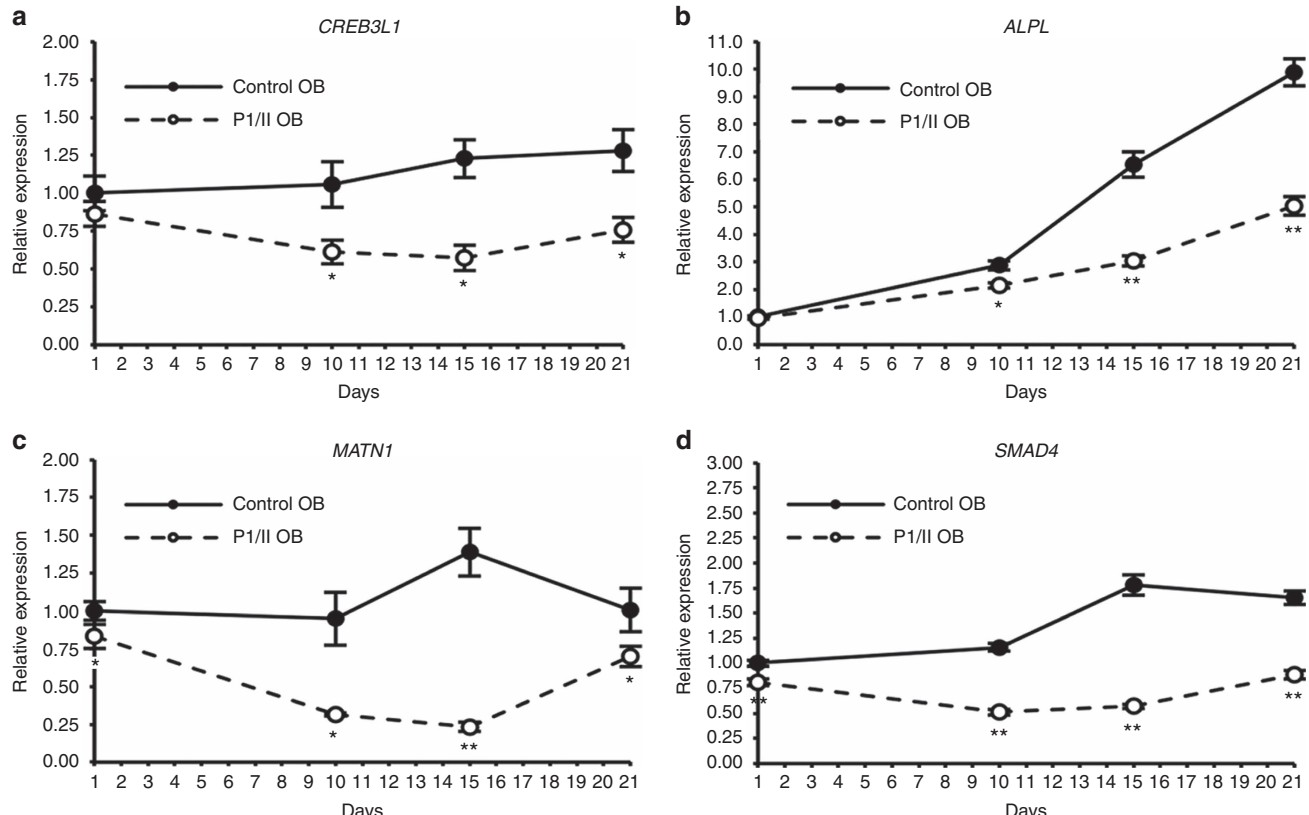

**Figure 6 | Effect of S2P deficiency on osteoblast differentiation. (a)** Transcripts encoding OASIS (*CREB3L1*), **(b)** bone-specific alkaline phosphatase (*ALPL*), **(c)** matrilin-1 (*MATN1*) and **(d)** Co-SMAD (*SMAD4*) are decreased in differentiating P1/II versus normal control osteoblasts in culture. *$P < 0.05$, **$P < 0.001$ by *t*-test. Error bars, s.d.

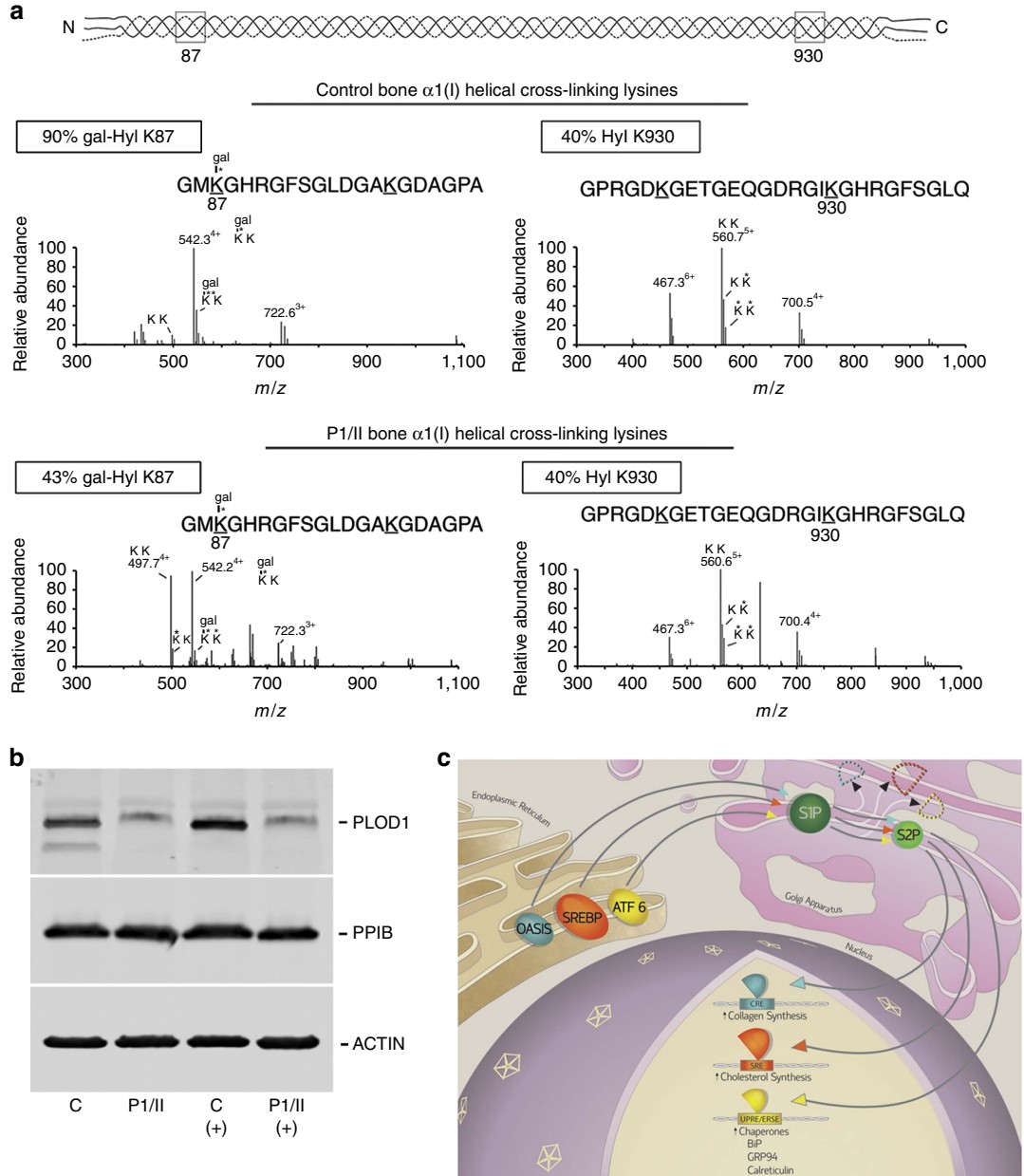

**Figure 7 | S2P deficiency causes abnormal post-translational modification of type I collagen. (a)** Mass spectrometric analysis of bacterial collagenase-digested peptides of control and P1/II bone type I collagen. Hydroxylation and consequent glycosylation of α1(I)K87 residues, involved in crosslink formation, is decreased in the proband sample by more than one-half, while hydroxylation of α1(I)K930 residues is normal, compared with normal control bone. K, lysine; K*, hydroxylysine; gal, galactosyl. **(b)** Immunoblots of lysates from control (C) and P1/II osteoblasts cultured in the absence or presence ( + ) of tunicamycin, an inhibitor of N-linked glycosylation that induces ER stress and OASIS cleavage. Despite normal levels of CyPB/PPIB, proband cells have significantly decreased LH1/PLOD1 protein, consistent with decreased hydroxylation of collagen lysyl residues (K87) involved in crosslink formation in extracellular matrix. **(c)** Schematic of RIP-mediated signalling. OASIS, SREBP and ATF6 translocate to the Golgi in response to ER stress or cholesterol depletion, where they are sequentially cleaved by S1P and S2P. The fully processed forms shuttle to the nucleus to activate transcription of genes involved in protein folding, lipid synthesis and extracellular matrix development[3,5,13,21].

normal (Fig. 7b)[12,16]. *PLOD1*/LH1 transcripts and protein were not consistently reduced in OI or IFAP/KFSD fibroblasts (Supplementary Fig. 7).

## Discussion

We identified an X-linked form of osteogenesis imperfecta in two independent pedigrees. Phenotypic inheritance pattern, linkage analysis and NGS were used to localize the causative gene in each family to *MBTPS2* at Xp22. The gene defects co-segregate with the phenotype in each pedigree and all affected individuals are the sons of obligate carriers. This gene identification extends the inheritance pattern of OI to include X-linked recessive. Notably, the OI phenotype of these *MBTPS2* defects is distinct from that of *PLS3* mutations, which cause an X-linked form of osteoporosis that is mostly apparent in middle-aged adults, but does cause fractures in some children[17]. Unlike *MBTPS2* defects reported here which cause moderate to severe OI, *PLS3* defects do not cause a generalized bone dysplasia, changes in bone shape or structure, or secondary features of OI.

*MBTPS2* encodes S2P, an integral membrane protein critical to RIP and essential for cholesterol metabolism[18]. Mutations previously reported to cause substitutions in S2P (Supplementary Fig. 8) underlie a spectrum of dermatological conditions: IFAP syndrome, characterized by a triad of ichthyosis follicularis, atrichia and photophobia[7], KFSD[9], BRESHECK syndrome[8] and Olmsted syndrome[19]. The affected individuals in our OI pedigrees do not have any symptoms associated with these conditions, even into adulthood. Both OI-causing substitutions in S2P are located in highly conserved intramembrane residues near the carboxyl end of the protein, and likely contribute to a specialized function. In Family I, the *MBTPS2* missense mutation results in a p.N459S substitution at the first residue of the S2P NPDG motif, which coordinates the zinc metal ion critical to protease catalytic function[10]. In addition, the serine substitution at this site introduces a potential O-GlcNAc glycosylation site (http://www.cbs.dtu.dk/services/YinOYang/), and addition of a bulky group might interfere with S2P interaction with its substrates. The Family II *MBTPS2* mutation results in S2P p.L505F. This substitution, in which leucine is replaced with the bulkier phenylalanine, is predicted to be pathogenic by multiple mutation analysis programs and may also interfere with S2P substrate interactions. Interestingly, the nearby p.N508S and p.N508T have a mild IFAP and KFSD phenotype[9], and most IFAP mutations are located adjacent to this critical region (p.R429-N508)[20].

S2P and S1P span the Golgi membrane and function sequentially in RIP to process crucial regulatory substrates transported from the ER membrane in times of ER-stress or sterol restriction, including OASIS, ATF6 and SREBP (Fig. 7c)[3,5,13,21]. The mature processed proteins then enter the nucleus where they activate pathways involved in cholesterol homoeostasis, unfolded protein response and bone formation. Our demonstration of S2P functioning in human bone development is supported by prior studies on defects of RIP components in animal models. Both S1P and S2P knockdown zebrafish showed abnormal cartilage development, independent of lipid defects[22]. Cartilage-specific S1P knockout mice (S1P$^{cko}$) have a lethal chondrodystrophy, with loss of hypertrophic chondrocytes and absent endochondral bone formation[23]. S1P$^{cko}$ chondrocytes also display differential downregulation of transcripts related to cholesterol and lipid biosynthesis[24]. In addition, ablation of the RIP substrate OASIS, a bZIP transcription factor, causes spontaneous fractures and severe osteopenia in mice[13] and homozygous deletion of *CREB3L1* (OASIS) causes recessive OI in one family[25].

The *MBTPS2* mutations causing X-linked OI do not impair *MBTPS2* expression or destabilize S2P protein, nor do *MBTPS2* mutations causing IFAP/KFSD. Instead, they disrupt S2P processing of RIP substrates, interfering with their downstream functions. First, both p.N459S and p.L505F fibroblasts and p.L505F osteoblasts display impaired OASIS cleavage. OASIS/*CREB3L1* expression is decreased in p.L505F osteoblasts, as is the expression of its nuclear binding target *SMAD4* (ref. 14), which, along with the reduced expression of osteoblast maturation marker alkaline phosphatase, indicates a broad defect in osteoblast differentiation. SMAD4 in turn co-activates transcription of matrix proteins such as matrilin-1 (ref. 15), consistent with decreased expression of *MATN1* in OI proband P1/II's osteoblasts. Second, type I collagen secretion is decreased in fibroblasts of all OI probands. The combination of normal collagen expression and reduced collagen secretion found in mutant *MBTPS2* fibroblasts differs from the expected decreased SMAD4 transcriptional activation of *Col1a1* (ref. 14), as well as from the murine *Oasis* knockout, in which reduced *Col1a1* expression was detected[13]. However, reduced collagen secretion

was not detected in fibroblasts of patients with mutant *CREB3L1* (OASIS)[25]. Furthermore, S1P$^{cko}$ chondrocytes retain type IIB procollagen due to a trafficking defect but do not exhibit decreased *Col2a1* expression[24]. Finally, the activation of RIP substrates ATF6 and SREBP during ER stress and sterol-free conditions, respectively, were significantly reduced in CHO-M19 cells transfected with reporter constructs containing mutant versus normal *MBTPS2*.

Notably, bone tissue from one X-OI proband revealed an underlying collagen-related defect. Type I collagen extracted directly from P1/II bone had significantly reduced hydroxylation of helical lysine residue 87 (K87) in both alpha chains, consistent with the proband's increased urinary LP/HP ratio. Although bone was not available from P1/I, his elevated urinary LP/HP ratio suggests a similar effect on collagen crosslinking occurs in his bone, while the LP/HP ratio was normal in two X-OI carriers and IFAP/KFSD patients (Supplementary Table 2). K87 is critical for collagen crosslinking in bone, which is a major contributor to bone strength[26]. Similarly reduced hydroxylation of type I collagen K87 to about half of wild-type control values occurs in a *Ppib*/CyPB knockout mouse model of type IX OI[12]. In the *Ppib* knockout mouse, the level of LH1/*PLOD1*, the enzyme which modifies K87, is normal, but its function is impaired by absence of its PPIase, CyPB. In contrast, LH1 protein is reduced in osteoblasts from X-linked OI proband P1/II by an unknown mechanism, although the level of CyPB is normal. Thus, the bone collagen defect in X-linked OI combines reduced collagen secretion into matrix and impaired collagen crosslinking. Disturbance of additional OASIS functions may contribute to the severe skeletal phenotype of X-linked OI, including apoptosis, downregulation of osteocalcin, osteopontin and bone sialoprotein, and ER stress, which overlaps ATF6 activation[27].

It is apparent that mutations at different positions in *MBTPS2* cause distinct syndromes, with no evidence at this time to suggest a spectrum disorder. A variety of skeletal malformations have been reported in IFAP-family syndromes, specifically vertebral malformation and cleft hands[28]. For example, the p.R429H IFAP mutation studied in our reporter assays causes vertebral abnormalities[7,8]. Short stature, inguinal hernia and microcephaly have also been noted in some IFAP patients. However, IFAP patients do not have the fracture susceptibility or generalized bone dysplasia that are the hallmarks of OI. However, the mechanism by which different mutations in *MBTPS2* cause totally different developmental disorders remains unclear. The biochemical studies comparing X-linked OI and IFAP fibroblasts show no clear distinction between phenotypes in cleavage of RIP substrates OASIS, ATF6 and SREBP. The major distinction between syndromes resides in the urinary crosslinks, with elevated LP/HP ratios in X-OI probands and not in X-OI carriers or an IFAP patient. This presents a reasonable hypothesis that impaired hydroxylation and crosslinking of bone collagen is the critical distinction for X-OI versus IFAP. Future studies elucidating the RIP pathways involved in bone development may reveal tissue-specific differences in collagen post-translational modification and crosslinking, or critical S2P bone-related substrates other than OASIS, ATF6 and SREBP. These studies of RIP pathways in bone development are of fundamental importance and will be facilitated by animal models and bone samples from additional patients with both X-OI and IFAP.

## Methods

**Detailed case reports.** *Family I.* The proband (P1/I; V7) of Family I, the Thai pedigree, was delivered at 39 weeks gestation by caesarean section due to multiple long bone fractures detected at 4 months gestation. His birth weight was 2,860 gm (10th centile), length 45 cm (<3rd centile), head circumference 34 cm (40th centile) and anterior fontanelle 5 × 5 cm. He had fractures of ribs, humeri, femora and tibiae (Fig. 2a). The proband received pamidronate starting at age 3 weeks.

He stood with support at 18 months. No additional fractures occurred through 24 months of age, when his weight was 10 kg (3rd centile), length 76.5 cm (50th centile for a 13-month-old boy) and head circumference 47.8 cm (40th centile). He had closed anterior fontanelle, positional plagiocephaly, blue sclerae, normal teeth, *pectus carinatum*, straight spine and upper extremity long bones but bowing of femora and tibiae. Karyotype showed normal 46, XY.

The pedigree contains 12 affected members, with seven living (Fig. 1a). Clinical assessments and blood samples were obtained from 18 individuals, including six affected (Supplementary Table 1). The proband's 32-year-old mother was healthy with normal weight, height and bone mineral density. The proband and his mother had normal serum calcium, phosphorus, alkaline phosphatase and lipid profiles. Affected adult males had fractures beginning during gestation, white sclerae, normal teeth, head circumference 52–58 cm (within mean ± 2.5 s.d. for Thai adults), variable scoliosis and pectal deformity, bowing of extremities with striking anterior angulation of tibiae, generalized osteopenia and final adult stature from 110 to 140 cm (50th centile for age 5–11 years).

*Family II.* Patient P1/II is a 26-year-old man (Fig. 2b and Supplementary Fig. 2a). The pregnancy was uneventful and he was delivered spontaneously at term. Birth weight was 2,970 g (10th–25th centile), with a length of 46 cm (3rd–10th centile). Direct postpartal bowing of the distal humeri and the proximal radii were indicative of osteogenesis imperfecta. Radiography showed bilateral fractures of the clavicles, several rib fractures and a consolidated prenatal fracture of the left femur (Supplementary Fig. 2b). Between age 14 months and 16 years, numerous fractures occured after only minor and moderate traumatic events, involving the femur 11 times (this stopped after the insertion of Bailey–Dubow elongation rods at age 8 years), the humerus 7 times, the lower arm 2 times and the tibia 3 times. Two fractures of the mandibulae occured after adequate traumatic events. There were no new fractures until age 24 years when one traumatic fracture of the lower leg occured. Treatment with 9 mg kg$^{-1}$ per day intravenous bisphosphonate was started at age 13 for 3 years. Current height at the age of 26 years is 110 cm (50th centile for 6-year-old male) and weight is 37 kg. Moderate kyphoscoliosis of the thoracic and lumbar spine is present. There are no signs of dentinogenesis imperfecta and the sclerae are white. An active wheelchair is mainly used. Unsupported standing is possible for a longer time; unsupported walking only a few steps, with support 5–10 m, and on a treadmill more than 1 km.

The maternal uncle P2/II is a 68-year-old man who was diagnosed with osteogenesis imperfecta quite soon after birth. Fracture rate and bending of the long bones were similar to P1/II. Since puberty, fractures are rarely observed unless through adequate trauma. He has a severe connatal *pectus excavatum*. His current height is 105 cm (50th centile for 5-year-old male) and his weight is 45 kg. Kyphoscoliosis of the thoracic and lumbar spine is more pronounced than in his nephew, and is rather progressive. He has no signs of dentinogenesis imperfecta and his sclerae are white. He can walk slowly with two crutches.

**Molecular genetic investigations of Family I.** *DNA from blood and dermal fibroblasts.* Proband and family whole blood and skin samples were collected after written informed consent was obtained under a protocol approved by the Institutional Review Board of the Faculty of Medicine, Chulalongkorn University. High-molecular-weight DNA was extracted from leukocytes of all the available family members, and from cultured skin fibroblasts of the proband, using the ArchivePure DNA Blood Kit (5 PRIME Inc., Gaithersburg, MD), according to the manufacturer's instructions.

*X-inactivation assay.* Eight female members (III1, III5, III7, IV5, IV13, IV18, IV21 and IV23) and one unaffected male (IV19) were subjected to X-inactivation analysis as previously described[29]. A total of 500 ng blood-derived DNA was digested with 10 U of the methylation-sensitive enzyme *Hpa*II at 37 °C for 16 h and enzyme-inactivated at 95 °C for 10 min. The first exon of the androgen receptor gene (*AR*) was amplified using a fluorescently labelled forward primer FAM-5′-CGCGAAGTGATCCAGAACCC-3′ and reverse primer 5′-GTTGCTGTT CCTCATCCAGG-3′ in digested and undigested DNA templates. The polymorphic CAG repeats in the *AR* gene were used to identify parental chromosomes. Methylation differences were analysed based on peak heights in the PCR products by Genemapper version 3.7 (Applied Biosystems, Foster City, CA).

*Linkage analysis.* Linkage analysis for Family I was performed according to the previously described methods[30]. X-chromosomal microsatellite markers were PCR-amplified using panel 28 of ABI Prism Linkage Mapping Sets-MD10 Version 2.5 (Applied Biosystems, Foster City, CA). An additional 21 polymorphic microsatellite markers were used for fine mapping on chromosome Xp22 between DXS8051 and DXS1068 (Supplementary Table 3). All fluorescently labelled primers were typed on an ABI Prism 3100 genetic analyzer (Applied Biosystems, Foster City, CA) with Genemapper version 3.7. For genetic mapping, the MLINK program was used to calculate two-point linkage analysis with the following model: X-linked recessive with high penetrance, disease and normal allele frequencies were set at 0.01 and 0.99. Maximum logarithm of the odds (LOD) score using the deduced genotyping of unavailable members was also shown in the bracket. Linkage analysis of the X-chromosome using a panel of microsatellite markers defined a 21.6-Mb critical region from markers DXS7108 to DXS1067, with a maximum LOD score of 3.31 (Supplementary Fig. 1a and Supplementary Table 3).

*Mutation analysis of candidate genes.* Candidate genes *PIR, TRAPPC2, PHEX, Cxorf15, AP1S2, CA5B, PDK3, PIGA, PRRG1, SMPX,* were selected because of their

inferred biological relevance for connective tissue functions. *HCCS, MSL3, PRPS2* on the telomeric end of the critical region were also studied. Primer pairs used for amplification of entire coding regions are available on request.

Sequencing of a set of genes involved in OI and bone fragility (*ALPL, BMP1, COL1A1, COL1A2, CREB3L1, CRTAP, FKBP10, IFITM5, LEPRE1, LRP5, PLOD2, PLS3, PPIB, SERPINF1, SERPINH1, SP7, TMEM38B* and *WNT1*) was performed (CTGT, Allentown, PA). Screening of IFITM5 by PCR and Sanger sequencing was normal. Sequencing of *COL1A2* revealed a single heterozygous IVS30 + 6T > C transition, which did not alter splicing (Supplementary Fig. 3). To determine the functionality of the change, a 260-bp fragment of genomic DNA from the OI proband, his mother, father and an affected male family member was PCR-amplified using 1.5 mM MgCl$_2$, 0.2 mM dNTPs, 200 nM forward (5′-AGCCT GTGTACTTATGCACT-3′) and reverse (5′-CAGTGGCTTTAAGGAGAAAG-3′) primers and 0.5 U Taq DNA polymerase (Life Technologies, Grand Island, NY) with 35 cycles of 94 °C for 30 s, 58 °C for 30 s and 72 °C for 30 s. To confirm that the *COL1A2* IVS30 + 6T > C transition was a nonfunctional genomic SNP, normal mRNA splicing was verified by RT–PCR analysis using total RNA from control and proband fibroblasts treated with or without emetine. After removal of contaminating DNA by treatment with DNA-free DNase (Applied Biosystems, Foster City, CA), poly-A mRNA was reverse transcribed using oligo d(T) and MuLV reverse transcriptase. The resulting cDNA was amplified from sequences corresponding to α2(I) exons 25–32 using 1 U Platinum Taq High Fidelity (Life Technologies, Grand Island, NY) and 1.5 mM MgSO$_4$, 0.4 mM dNTPs, 15 nmol forward (5′-CATTGGATTCCCTGGACCCAAAGGCCCCAC-3′) and reverse (5′-GGAGTCCACTAGGACCAGATGGACCAGCAG-3′) primers, and 35 cycles of 94 °C for 1 min, 67 °C for 20 s and 72 °C for 45 s. PCR products were electrophoresed on 6% polyacrylamide and visualized by ethidium bromide staining.

*Comparative Genomic Hybridization.* CGH was performed on test DNA (affected male, IV17) and reference DNA (unaffected male, IV10) for the determination of copy number variants (Macrogen Inc., Seoul, Korea). DNAs were independently labelled with fluorescent dyes, co-hybridized to a NimbleGen Human CGH 385K chromosome X Tiling array, and scanned using a 2-μm scanner. Log2-ratio values of the probe signal intensities (Cy3/Cy5) were calculated and plotted versus genomic position using Roche NimbleGen NimbleScan software. Data are displayed in Roche NimbleGen SignalMap software. CGH did not show any imbalances in the critical region.

*Next-generation sequencing.* We performed targeted resequencing by NGS of the 18.6-MB linkage region between DXS1224 and DXS1067 interval on affected male (IV17) and unaffected male (IV10) genomic DNAs (Macrogen Inc., Seoul, Korea). DNA was captured on a customized NimbleGen 2.1 array (Roche NimbleGen, Madison, WI) with a capturing capacity of 33 Mb. The targeted region corresponded to positions 13,235,460 to 31,798,001 bp on Chromosome Xp22 according to the UCSC hg19 Assembly. Capture efficiency varied across the target, with a mean read depth of 121.1X. The captured library was subsequently sequenced using the Illumina platform Genome Analyzer II X (GAIIX) in a single-end 76 bp configuration. Sequence reads were mapped against UCSC hg19 using BWA software (http://bio-bwa.sourceforge.net/). The SNPs and Indels were detected by SAMTOOLS (http://samtools.sourceforge.net/) and annotated by SIFT (http://sift.jcvi.org/) (Supplementary Table 4). We found 25 variants in coding sequences. After excluding known SNPs in dbSNP Build 135, 1000 Genomes and HAPMAP, and variants present in the unaffected relative, two candidates remained: c.1376A > G (p.N459S) in *MBTPS2* and c.532G > A (p.V178M) in *FAM48B1*.

*Mutation confirmation and segregation studies.* PCR amplification and Sanger sequencing were performed to confirm the *MBTPS2* mutation in the proband and his parents. Primer pairs and PCR conditions for the amplification of the coding exons of *MBTPS2* including the intron–exon boundaries are available on request. Restriction enzyme digestion with *Bsg*I was used for cosegregation study and normal control screening. The identified mutation was analysed with ClustalX for conservation analysis. Alamut, Polyphen2, SIFT, Panther and align GVGD were used for prediction of mutation function.

**Molecular genetic investigations of Family II.** *Sequencing of osteogenesis imperfecta candidate genes.* Exclusion of disease-causing mutations segregating with the phenotype was performed for *ALPL, BMP1, COL1A1, COL1A2, CREB3L1, CRTAP, FKBP10, IFITM5, LEPRE1, LRP5, PLOD2, PLS3, PPIB, SERPINF1, SER-PINH1, SP7, TMEM38B* and *WNT1* (CTGT, Allentown, PA). A heterozygous c.4018G > A (p.G1340S) in *COL1A1*, reported as rs147936946 in the dbSNP database was found in P1/II and his healthy father. This variant was absent in the affected maternal uncle (P2/II), as well as in the proband's mother (Supplementary Fig. 3c).

*Genome-wide linkage analysis.* For the five living members of Family II (Supplementary Fig. 1) a genome-wide linkage scan was performed using the HumanCytoSNP-12v2 BeadChip (Illumina, San Diego, CA) according to the supplier's instructions. Raw SNP call data were processed with the Genotyping Analysis Module in GenomeStudio 1.6.3 (Illumina, San Diego, CA). Homozygosity mapping was performed with the Allegro program, under the assumption of a recessive mode of inheritance in absence of consanguinity loops.

*X-inactivation assay.* X-inactivation analysis was performed on gDNA from the mother of P1/II according to Beever *et al*[31]. PCR products were analysed on an ABI

3130 Capillary DNA Sequencer (Applied Biosystems) and the methylation status was analysed by Genemapper version 4.0 (Applied Biosystems) based on the peak heights of the PCR products.

*Exome enrichment and high-throughput sequencing.* Genomic DNA was extracted from whole blood using the MagNA Pure Compact System (Roche Life Science, Indianapolis, IN, USA). DNA library preparation was performed with the TruSeq DNA Sample Preparation Kit (Illumina, San Diego, CA) according to the manual instructions. Enrichment of X-chromosomal genes was done by using a custom designed NimbleGen Sequence Capture Microarray (solid array with 6962 targets on the X chromosome or 1.4 Mb). The captured library was sequenced on a MiSeq sequencer (Illumina, v1) with $2 \times 150$ cycles of paired-end sequencing.

*Data analysis.* Raw data processing, sequence read alignment from FASTQ to BAM format and variant calling to generate VCF files were performed with the MiSeqReporter Software (Illumina, San Diego, CA). Aligned BAM files with removed duplicated reads were further technically analysed using the SeqMonk program version v0.25.0 (http://www.bioinformatics.babraham.ac.uk/projects/seqmonk). Variant filtering was performed by GeneTalk (https://www.gene-talk.de/).

*Mutation confirmation and segregation study.* Mutation screening of *MBTPS2* and segregation analysis of the mutation within the family was performed by fluorescent bidirectional sequencing of genomic amplified PCR products on an ABI 3100 automated sequence detection system (Applied Biosystems, Foster City, CA, USA). PCR conditions and primer sequences are available from the authors on request.

Bioinformatics analysis aimed to predict pathogenicity of the identified L505F variant was performed with Provean, Panther, MutationTaster and PolyPhen2. Conservation analysis of the protein variant was performed by multiple sequence alignment of *MBTPS2* with ClustalX, ClustalW and BOXSHADE programs (Fig. 2e).

*Urinary pyridinoline analysis.* Total urinary pyridinoline crosslinks, LP and HP, were measured in the affected individuals and obligate carriers of Family I and Family II, as well as in an *MBTPS2* R429H IFAP patient and unaffected carrier (Supplementary Table 2) by HPLC as described[32].

**Luciferase assay.** CHO-M19 cells, a deleted *Mbtps2* orthologous cell line, were used for transfection of reporter constructs. Cells were maintained in Ham's F-12 medium supplemented with 10% fetal bovine serum at $37\,°C$ with 5% $CO_2$. We performed luciferase assays by transient transfection of CHO-M19 cells either with p5xATF6 or pSRE reporters as previously described[5,7]. The p5xATF6 and pSRE reporters contain either an ER-stress response element or a sterol regulatory element, respectively, two elements that are known targets of S2P. These assays are an indirect measurement of the transcriptional activity of S2P cleavage products on these two elements. For the p5xATF6 reporter, cells were set up at a density of $1.5 \times 10^5$ cells per well in triplicate in 12-well culture plates. Cells were transfected with 1 μg of expression plasmid containing normal, mutant (N459S, R429H and L505F) *MBTPS2* or no insert, together with 800 ng of p5xATF6-GL3 and 50 ng of pRL-SV40 as a transfection control. The total amount of DNA was adjusted to 2 μg per dish by addition of pdEYFP-C1amp. At 24 h post-transfection, cells were treated with $2\,\mu g\,ml^{-1}$ of tunicamycin dissolved in DMSO to induce stress. Untreated cells received 0.2% DMSO. After incubation for 24 h, the firefly and renilla luciferase activities were measured using a dual luciferase assay system according to the manufacturer's protocol (E1910 Promega kit, Madison, WI). For the pSRE-reporter, cells were set up at a density of $1.5 \times 10^5$ cells per well in triplicate in a 12-well plate. On day 2, the cells were switched to medium containing 1:1 Ham's F12:DMEM, 5% lipoprotein-deficient serum and 20 mM sodium oleate for 7 h. The transfected plasmids were either wild-type S2P or one of the three mutants (p.N459S, p.R429H and p.H171F) and the pSRE reporter. Cells were then switched to medium containing 1:1 Ham's F12:DMEM, 5% lipoprotein-deficient serum, 50 mM compactin and 50 mM mevalonate supplemented with sterol ($10\,mg\,ml^{-1}$ cholesterol, $1\,mg\,ml^{-1}$ 25-hydroxycholesterol). Untreated cells were treated with 0.2% ethanol (0.2% w/v). The firefly and renilla luciferase activities were measured after 16 h of incubation. The ratio between firefly luciferase and renilla luciferase luminescence from different constructs and control was analysed by one-way analysis of variance (ANOVA). Specific pairwise comparison between luciferase luminescence in each group used the least significant difference test. One-way ANOVA and least significant difference were included in SPSS software version 22.

**Western blot analysis.** Cell lysates from primary fibroblasts and osteoblasts were collected in RIPA buffer supplemented with a protease inhibitor cocktail (Sigma-Aldrich, St Louis, MO), and used for western blotting of MBTPS2 (Cell Signaling Technology, Danvers, MA), OASIS (R&D Systems, Minneapolis, MN) and PLOD1/LH1 (Abgent Inc., San Diego, CA) proteins. Cells were grown to confluence and treated with tunicamycin ($2\,\mu g\,ml^{-1}$), thapsigargin (1 mM) and/or ALLN ($25\,\mu g\,ml^{-1}$) for 2 h before harvest. Protein lysate (50 μg) was loaded per lane on a 4–20% (MBTPS2, PLOD1) or 10% (OASIS) Tris-Glycine Ready gel (Bio-Rad, Hercules, CA), blotted onto 0.2 μm nitrocellulose and blocked with Odyssey blocking buffer (MBTPS2) or 5% BSA plus $1\times$ casein (OASIS, PLOD1) before probing with antibody overnight. Blots were washed, incubated with secondary IR-conjugated antibodies for 1 h, washed and visualized on a LI-COR Odyssey infrared imager (LI-COR, Lincoln, NE).

**Collagen biochemical studies.** Dermal fibroblast cultures were established from skin punch biopsies and grown in Dulbecco's Modified Eagle Medium (DMEM) containing 10% fetal bovine serum (FBS), $100\,U\,ml^{-1}$ penicillin and $100\,\mu g\,ml^{-1}$ streptomycin.

For type I collagen biochemical studies, normal control and proband primary fibroblast and osteoblast (see below) cultures were labelled for 16 h in serum-free DMEM containing $437.5\,\mu Ci\,ml^{-1}$ L-[2,3,4,5-³H]-proline. Procollagens were precipitated with ammonium sulfate, pepsin digested and electrophoresed on 6% SDS-urea polyacrylamide gels under nonreducing conditions[33]. Procollagen secretion by control and proband fibroblasts (Fig. 5b) was determined in 24-h conditioned media using the Sircol collagen assay (Biocolor Ltd, Carrickfergus, UK) according to the manufacturer's specifications. Collagen protease sensitivity was analysed as previously described[34]. For protease sensitivity assays, type I collagen from the medium of cultured proband fibroblasts labelled overnight with $437.5\,\mu Ci\,ml^{-1}$ L-[2,3,4,5-³H]-proline was digested for 1, 3 and 5 min with trypsin and chymotrypsin (100 and $250\,mg\,ml^{-1}$ final concentration, respectively) at $37\,°C$ without prior cooling. Digests were stopped with $1\,mg\,ml^{-1}$ soybean trypsin inhibitor and analysed by SDS–urea PAGE.

Differential scanning calorimetry (DSC) was performed in buffer containing 0.2 M sodium phosphate and 0.5 M glycerol at pH 7.4, from 10 to $55\,°C$ in a Nano III DSC instrument (Calorimetry Sciences Corporation, Lindon, UT), as previously described[35]. Amino-acid composition was analysed by high pressure liquid chromatography (AIBiotech, Richmond, VA).

Collagen extracellular matrix deposition was analysed as previously described[36]. In brief, post-confluent proband and normal control fibroblasts were cultured for 2 weeks in DMEM containing 10% FBS with $100\,\mu g\,ml^{-1}$ ascorbic acid. After 14 days, fibroblasts were labelled 24 h with $406.25\,\mu Ci\,ml^{-1}$ L-[2,3,4,5-³H]-proline and the media and extracellular matrix were harvested. Collagens were sequentially extracted from the matrix with 150 mM NaCl, 0.5 N acetic acid and then $50\,\mu g\,ml^{-1}$ pepsin.

**Expression studies.** Total fibroblast RNA was extracted from primary fibroblasts using TriReagent (Molecular Research Center, Cincinnati, OH), followed by DNase treatment and RNA integrity verification on an Agilent 2100 Bioanalyzer (Agilent, Santa Clara, CA). Complementary DNA was reverse transcribed from 5 μg RNA using a High-Capacity cDNA Archive Kit (Life Technologies Corporation, Carlsbad, CA). Taqman Gene Expression Assays were used to determine the transcript level of *MBTPS2* (Hs00210639_m1), *COL1A1* (Hs00164004_m1), *COL1A2* (Hs00164099_m1), *PLOD1* (Hs00609368_m1), *CREB3L1* (Hs00223565_m1), *ALPL* (Hs01029144_m1), *MATN1* (Hs00159075_m1) and *SMAD4* (Hs00929647_m1) (Life Technologies Corporation, Carlsbad, CA). Relative expression of each gene of interest was compared with the expression of *GAPDH* (Hs99999905_m1) and normalized to normal control primary fibroblasts.

**Osteoblast studies.** Osteoblast primary cultures were established from surgical bone chips and cultured in αMEM (Life Technologies, Grand Island, NY) supplemented with 10% FBS, penicillin and streptomycin at $37\,°C$ and 8% $CO_2$ (ref. 37). Only passages 1 and 2 were used for experiments. For differentiation, osteoblast cultures were grown to confluence and treated with differentiation medium (αMEM containing 10% FBS, $25\,\mu g\,ml^{-1}$ L-ascorbic acid, dexamethasone ($10^{-8}$ M), and 2.5 mM 2-glycerophosphate for 3 weeks.

**Mass spectral analysis of bone tissue collagen.** Demineralized bone was digested with bacterial collagenase and the resulting collagen-derived peptides were separated by reverse-phase HPLC, as previously described[32]. Electrospray MS was performed on in-gel trypsin digests and individual HPLC column fractions using an LTQ XL ion-trap mass spectrometer equipped with in-line liquid chromatography (Thermo Scientific, Waltham, MA) via a C4 5-μm capillary column ($300 \times 150\,mm$; Higgins Analytical RS-15M3-W045) eluted at $4.5\,\mu l\,min^{-1}$. The LC mobile phase consisted of buffer A (0.1% formic acid in MilliQ water) and buffer B (0.1% formic acid in 3:1 acetonitrile:n-propanol v/v). The LC sample stream was introduced into the mass spectrometer by electrospray ionization with a spray voltage of 4 kV. Proteome Discoverer search software (Thermo Scientific) was used for peptide identification using the NCBI protein database. Proline and lysine modifications were examined manually by scrolling or averaging the full scan over several minutes so that all of the post-translational variations of a given peptide appeared together in the full scan.

**Study oversight.** This study was conducted according to the Declaration of Helsinki for Human Rights and approved by Swiss Ethics and by the Institutional Review Board of the Faculty of Medicine, Chulalongkorn University. Written informed consent was obtained from the patients or their parents.

**Data availability.** The data that support the findings of this study are available from the corresponding author on request.

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

## Acknowledgements

We thank the patients and their families for participating in this study and supporting our work. Osteoblast cultures were kindly established by Professor Thomas Braulke, University Medical Center Hamburg-Eppendorf, Children's Hospital, Hamburg, Germany. CHO-M19, EYFP-MBTPS2, pSRE-GL4.23 and pRL-SV40 constructs were kindly provided by Professor Karl-Heinz Grzeschik, Department of Human Genetics, Philipps-Universität, Marburg, Germany. IFAP fibroblast cultures were generously provided by Dr Cynthia J. Tifft, Pediatric Undiagnosed Diseases Program, NHGRI, NIH. KFSD fibroblast cultures were kindly provided by Dr Emmelien Aten, Department of Human and Clinical Genetics, LUMC, Leiden, The Netherlands. The p5xATF6-GL3 was kindly provided by Ron Prywes of Department of Biological Sciences, Columbia University, New York, USA. We thank Angelika Schwarze and Anke Jeschke for expert technical assistance, and Professor Marius Kraenzlin for urinary pyridinoline measurement. This study was supported by the Thailand Research Fund (RTA56800003 and TRG5480013) and its the Chulalongkorn Academic Advancement into its 2nd century project to V.S., by the Swiss National Research Foundation (SNF Grant No. 310030_138288) to C.G. and M.R., by NIH grants AR037318 (NIAMS) and HD070394 (NICHD) to D.R.E., and by NICHD Intramural Funds to J.C.M. and S.L. The illustration in Fig. 6c was generated by Yumiko Shepherd, Unit of Computer Support Services, NICHD.

## Author contributions

S.A. and P.Y. performed X-inactivation studies, linkage analysis, analysed NGS and CGH data and performed functional experiments of the identified mutation for Family I. Also for Family I, C.S. prepared samples, and did PCR analyses of collagen and gene sequencing; S.T. conducted reporter assays and functional studies. For Family II, I.K. ascertained the family and elaborated the clinical data. U.L. performed collagen biochemical analyses, expression and functional studies of MBTPS2. A.J. and B.K. performed linkage analyses, and K.L. and B.R. did X-inactivation studies and NGS. E.M. conducted DSC under the supervision of S.L. M.W. and D.E. conducted mass spectroscopy analyses of P1/II bone sample. A.M.B. produced immunoblots of MBTPS2, PLOD1 and OASIS cleavage. W.A.C. performed collagen biochemical analyses, expression quantitation of *MBTPS2* and *PLOD1* and studies of cultured osteoblasts. Patient samples were collected by K.S., V.S. (Family I) and I.K., M.R. and C.G. (Family II). J.C.M., K.S., V.S., M.R. and C.G. designed and supervised the study. J.C.M. drafted the manuscript with significant input from K.S., V.S., W.A.C., U.L. and C.G. All authors read and approved the final manuscript.

## Additional information

**Competing financial interests:** The authors declare no competing financial interests.

