## [Peer Review File · Nature Communications]

Reviewers' Comments:

Reviewer #1 (Remarks to the Author)

This manuscript describes novel mutations in MBTPS2 encoding site-2 metalloprotease (S2P) in two kindred with X-linked osteogenesis imperfecta (OI), thus providing a new OI locus.

Mutations in S2P have been previously associated with dermatological conditions such as IFAP and KFSD, but not with classical OI (bone deformities and recurrent fractures), hence the claims of the paper are novel and of clinical relevance.

The authors performed functional analysis in patient cells, including OI and IFAP/KFSD fibroblasts, in order to determine the molecular basis of S2P-OI. They demonstrate that the two newly identified OI mutations, like the IFAP and KFSD mutations, impair regulated intramembrane proteolysis (RIP) of transcription factors OASIS and ATF6.

Additionally, they analyzed type I collagen biochemistry and detected a number of molecular defects, some of which are not fully shared by both OI pedigrees. The Thai OI mutation (family I; p.N459S), which involves a highly conserved amino acid required for ion coordination (Rudner DZ et al. PNAS 1999), and the IFAP/KFSD mutations, all result in both decreased COL1A1 expression and type I collagen secretion. The p.L505F OI mutation leads to reduced type I collagen secretion, but does not affect COL1A1 expression, perhaps suggesting that this is a hypomorphic mutation.

The authors also performed studies in p.L505F osteoblasts and demonstrate in this proband impaired osteoblast development and decreased K87 lysyl hydroxylation as a result of lower LH1 protein levels specifically in bone.

The paper is clear and well-written and I support publication after addressing a few comments, which according to my opinion, require attention by the authors.

1) Variations in K87 lysyl hydroxylation and lower levels of LH1 have only been clearly shown in family II (p.L505F). It would be ideal if bone tissue from affected individuals of family I could also be analyzed. However, I understand that this is not always possible and it rises ethical issues. Therefore the authors should include at least a sentence discussing whether they think this is a common pathomechanism for S2P-OI or it may depend on the mutation. Interestingly urinary LP/HP ratio appears to be more increased in family II than I (supplementary table 2).

2) All figures are missing indications for statistical significance resulting difficult to determine whether some of the differences are statistically significant. Just two examples:

-Figure 3B. Is the reduction in luciferase activity of L505F with respect to the control statistically significant?

-Supplementary figure 8. Are the differences in the expression levels of PLOD1 in fibroblast lines

statistically significant?.

3) Material and methods (Supplementary material): "We performed luciferase assays by transient transfection of CHO-M19 cells either with p5xATF6 or pSRE reporters as previously described". pSRE reporters are also mentioned in acknowledgements.

I understand that only data referring to the ATF6 reporter is provided in the manuscript. Was the effect of S2P-OI mutations on SREBP factors tested?. If so, it would be a valuable information to add.

4) The authors claim that "collagen deposited into matrix in culture by S2P p.N459S fibroblasts had a decreased proportion of collagen containing mature cross-links". Can they discuss the reason that could be behind this alteration?

Minor comments:

-Results. Mutation identification. For family I, 644 controls were analyzed. Were all of them ethnically matched controls?. Please indicate whether the S2P-OI mutations identified are present or not in the current databases of genetic variants (EXAC, EVS).

-Segregation analysis for p.L505F. Please indicate more precisely all the individuals that were actually analyzed.

-Since genomic nucleotide positions are given, please indicate the human genome build used as reference. Equally indicate the reference sequence used to name MBTPS2 mutations.

-As the probands of both families are not exactly equal, I suggest to pay attention when using this word. It would be more accurate to specify "proband I" or "proband II" when a particular feature does not apply or has not been demonstrated in both families.

Reviewer #2 (Remarks to the Author)

The interesting manuscript „MBTPS2 mutations cause defective regulated intramembrane proteolysis in X-linked osteogenesis imperfecta" by Lindert et al. describes the association of specific missense mutations in MBTPS2, coding for S2P, with an X-linked recessive form of osteogenesis imperfecta (OI).

MBTPS2 missense mutations impairing intramembrane protein cleavage by S2P previously had been implicated in IFAP syndrome and phenotypically related dermatological developmental disorders.

Therefore, the detection of MBTPS2 mutations causing X-linked OI is highly interesting.

These mutations affect the protein at intramembrane amino acid positions which are in close neighborhood to the ones causing dermatological phenotypes of the IFAP spectrum. The obvious question, which the authors attempt to tackle, addresses the mechanism by which adjacent mutations in the catalytically active center cause totally different developmental

disorders.

Towards this end, the authors analyze the effect of MBTPS2 mutations associated with OI, IFAP syndrome, and KFSD on the function of the transcription factors ATF6 and OASIS which result from cleavage by S2P.

The results of these interesting and well documented functional studies, unfortunately, are insufficient to shed light on the critical question of the contrasting phenotypic effects of these mutations:

In the experiments where cells from OI patients are compared in parallel with cells from IFAP, KFSD, and wild-type, no significant differences show up between the mutants.

Differences between cells of one OI mutant only, in particular osteoblasts, and wild-type cells hint at a deleterious effect of this OI mutation on collagen synthesis, collagen post-translational modification and crosslinking as well as on osteoblast differentiation. However, it has not been tested, if a similar effect might be demonstrated in osteoblasts with the other OI-mutation nor the IFAP and KFSD mutations in MBTPS2.

Accordingly, the conclusion in the "Discussion" chapter: "Whilst X-linked OI and IFAP fibroblasts share the biochemical abnormality of impaired OASIS cleavage, the distinction between these syndromes is likely to lie in tissue specific differences in substrate cleavage and collagen post-translational modification and crosslinking" sounds vague and seems not to be supported by the data.

From the results, one might rather suspect, that OASIS and, eventually, ATF6 are not the crucial substrates affected by the OI-specific MBTPS2 mutations.

Major suggestions:

1. The last part of the discussion, dealing with the mechanism leading to OI versus phenotypes of the IFAP complex, might be clarified to differentiate between conclusions supported by the results and speculations calling for further studies.
2. In the Methods chapter, the authors describe Luciferase experiments with p5xATF6 or pSRE reporters. Results with pSRE are not shown, but might be interesting in view of the question if the S2P substrates ATF6 and OASIS employed in this research are the critical ones leading to X-linked OI.
3. Page 22: Legend to Fig. 6c: References for the results, on which this schematic is based, should be given.
4. Supp. Fig. 1 and Fig. 5b: Individuals V5 and IV21 are described as possible carriers (see Fig. 5b). IV5 does not have the haplotype of mutated chromosome. IV21 is shown in Supp.Fig. 1 as carrier, based on the X-inactivation result only.

Minor:

Page 3: Reduced collagen crosslinks presumptively undermines bone strength....

→..undermine...

Page 8: ...predict both MBTPS2 variants p.N459S and p.L505F as pathogenic... (Do the prediction programs determine the mutations as pathogenic or possibly pathogenic?)

Page 12: dito!is predicted to be pathological by multiple mutation analysis programs...

→...possibly pathogenic...

Page 17: Reference Naiki et al. is incomplete.

Page 22: Legend to Fig. 6c: References for the results, on which this schematic is based, should be given.

Supp. Page 9: ...according to the manual instructions...→ ...to the suppliers instructions...

Supp page 17: ... The MBTPS2 c.1376A>G (p.N459S) mutation segregated with phenotype in all members. →...all affected members?...

Supp. Page 21: ...(b) Levels of X-inactivation of female members were shown.... →..are shown...

Supp. Pages 22 and 24: Supp Figs 6 and 8. It may be helpful to explain "NL" in the legend (Normal).

Supp. Page 33: Supp. Table 6: According to "Methods" luciferase assays have been performed for p5xATF6 or pSRE reporters. What is shown here? A bit more of legend would be helpful.

If it shows only the result of an assay with p5xATF6, where is the result of pSRE shown?

Supp.Pag3 34: Ref.2 is incomplete.

Reviewers' Comments:

Reviewer #1 (Remarks to the Author)

In the revised manuscript and the itemized response to reviewers, the authors have taken account of comments and have performed (or added) new experiments including statistical analysis ,LP/HP ratio of additional individuals and SRE Luciferase Reporter Assays. I have no further comments and I recommend the article for publication

Reviewer #2 (Remarks to the Author)

In their revised version of the interesting and well written manuscript „MBTPS2 mutations cause defective regulated intramembrane proteolysis in X-linked osteogenesis imperfecta" Lindert et al. have completely addressed the concerns and comments raised in the review of the original version. In particular, they show now additional data clarifying the impact of the X-linked OI-associated MBTPS2 mutations in luciferase experiments with pSRE reporters and interpret the critical role of the S2P substrates ATF6 and OASIS for causing X-linked OI more cautiously.

One minor discrepancy now came to my attention in the manuscript. On page 7 the authors report the skewed pattern of X-inactivation of four obligate carriers of Family I, whereas Supp. Fig. 5 lists five obligate carriers. The clarification of this issue does not require another review process but could be clarified during the further process of publication.

Thus, I support publication of the revised version.

We were pleased that our manuscript by Lindert, Cabral, Ausavarat, *et al.*, "MBTPS2 mutations cause defective regulated intramembrane proteolysis in X-linked osteogenesis imperfecta" was found to be interesting and convincing by the Referees. We appreciate their comments to improve our manuscript. The response to each of their suggestions is detailed below.

Response to Reviewer #1:

Major comments:

- 1) Variations in K87 lysyl hydroxylation and lower levels of LH1 have only been clearly shown in family II (p.L505F). It would be ideal if bone tissue from affected individuals of family I could also be analyzed. However, I understand that this is not always possible and it rises ethical issues. Therefore the authors should include at least a sentence discussing whether they think this is a common pathomechanism for S2P-OI or it may depends on the mutation. Interestingly urinary LP/HP ratio appears to be more increased in family II than I (supplementary table 2).

We agree that it would be ideal to analyze bone tissue with both mutations, but the proband from family I (N459S) has not required orthopedic surgery and his parents have not consented to an elective biopsy.

The urinary LP/HP ratio does provide an indirect insight into bone crosslinks. The urinary LP/HP ratio reported for P1/I in the original submission was obtained at age 3 yrs old and was marginally increased vs controls, raising the issue of whether the two mutations have a common pathomechanism. In growing children (as in the index patient of Family I) the contribution of collagen type II to urinary pyridinoline is high, compared to adult patients where the type II contribution is negligible. Since collagen type II is more hydroxylated than type I, the contribution of underhydroxylated collagen I to the urinary pyridinoline ratio is somewhat obscured in children by the natural hyperhydroxylation of collagen type II.

We obtained an additional urine from P1/I at age 6.5 years, and from an affected adult (IV17/I) and an adult carrier (IV18/I) in Family 1. Supplemental Table 2 now shows that affected individuals in both Families have increased LP/HP ratio, while carriers and an IFAP patient do not. This data is described in Results on p.7. In the revised Discussion on pp.14-15, we incorporate this data into a proposed common pathomechanism for X-OI in the two pedigrees.

- 2) All figures are missing indications for statistical significance resulting difficult to determine whether some of the differences are statistically significant. Just two examples:
-Figure 3B. Is the reduction in luciferase activity of L505F with respect to the control statistically significant?

-Supplementary figure 8. Are the differences in the expression levels of PLOD1 in fibroblast

lines statistically significant?.

Indications for statistical significance have been added to the Figures. For Figure 3b, all mutations show significant reduction in ATF6 luciferase activity with respect to control, and p values are now indicated in the Figure Legend on page 22. For Figure 4a, collagen expression is significantly decreased in P1/I and IFAP/KFSD, but not in the L505F fibroblasts and osteoblasts, although collagen secretion is significantly decreased in all *MBTPS2* mutant cells (Figure 4b), as noted on p.22.

In Figure 8, PLOD1 expression is significantly down in untreated fibroblasts only in P1/I and IFAP/KFSD, but after tunicamycin treatment PLOD1 is significantly decreased in P1/II as well.

3) Material and methods (Supplementary material): "We performed luciferase assays by transient transfection of CHO-M19 cells either with p5xATF6 or pSRE reporters as previously described". pSRE reporters are also mentioned in acknowledgements.

I understand that only data referring to the ATF6 reporter is provided in the manuscript. Was the effect of S2P-OI mutations on SREBP factors tested?. If so, it would be a valuable information to add.

Yes, we did study luciferase assays with pSRE reporters, but only for the Thai mutation, in part because it was difficult to obtain critical reagents, especially sterol-depleted culture media, when we later added the Swiss mutation. We now show the pSRE results for the Thai mutation as Figure 3c, and the detailed data is in Supplementary Table 8. In the absence of sterol, the Thai OI mutation had reduced luciferase response compared to normal S2P, but higher than the luciferase activity with the IFAP mutations (p.R429H and p.H171F). We have also added a paragraph to the Material and Methods on Supplementary p. 11 describing the method for this assay.

4) The authors claim that "collagen deposited into matrix in culture by S2P p.N459S fibroblasts had a decreased proportion of collagen containing mature cross-links". Can they discuss the reason that could be behind this alteration?

We have now added a sentence to p. 10, indicating that the mature crosslinks are derived from the K87 hydroxylation, so the decrease in cross-links suggests that this site may be underhydroxylated, despite normal LH1 levels in fibroblasts by Western blot. This fibroblast collagen data provides indirect support for a similar interpretation of the collagen biochemistry results in L505F bone.

Minor comments:

-Results. Mutation identification. For family I, 644 controls were analyzed. Were all of them ethnically matched controls?. Please indicate whether the S2P-OI mutations identified are present or not in the current databases of genetic variants (EXAC, EVS).

Yes, all 644 control X chromosomes (181 females + 282 males) for the Thai OI mutation are Thai, now indicated on page 7. Both mutations are not identified as variants by current databases including ExAC, EVS, 1000G, and dbSNP build 144 databases (accessed Feb 18, 2016), as now indicated on page 8 in Results and Supplemental Table 6.

-Segregation analysis for p.L505F. Please indicate more precisely all the individuals that were actually analyzed.

The segregation analysis was done on all family members still alive at time of sampling, that is all but individuals I/1, I/2 and II/4. This is now indicated in the Legend to Supplemental Figure S2.

-Since genomic nucleotide positions are given, please indicate the human genome build used as reference. Equally indicate the reference sequence used to name MBTPS2 mutations.

We use Genome Reference Consortium Build 38 (GRCh38) as reference for the positions of NGS results, and NCBI Reference Sequences for transcript (NM_015884.3) and protein (NP_056968.1) identification. This is now indicated on p. 7 of Results.

-As the probands of both families are not exactly equal, ...it would be more accurate to specify "proband I" or "proband II" when a particular feature does not apply or has not been demonstrated in both families.

We have made this adjustment throughout the text.

Reviewer #2 (Remarks to the Author):

The obvious question, which the authors attempt to tackle, addresses the mechanism by which adjacent mutations in the catalytically active center cause totally different developmental disorders.

Towards this end, the authors analyze the effect of MBTPS2 mutations associated with OI, IFAP syndrome, and KFSD on the function of the transcription factors ATF6 and OASIS which result from cleavage by S2P. The results of these interesting and well documented functional studies, unfortunately, are insufficient to shed light on the critical question of the contrasting phenotypic effects of these mutations: In the experiments where cells from OI patients are compared in parallel with cells from IFAP, KFSD, and wild-type, no significant differences show up between the mutants.

Differences between cells of one OI mutant only, in particular osteoblasts, and wild-type cells hint at a deleterious effect of this OI mutation on collagen synthesis, collagen post-translational modification and crosslinking as well as on osteoblast differentiation. However, it has not been tested, if a similar effect might be demonstrated in osteoblasts with the other OI-mutation nor the IFAP and KFSD mutations in MBTPS2.

Accordingly, the conclusion in the "Discussion" chapter: "Whilst X-linked OI and IFAP fibroblasts share the biochemical abnormality of impaired OASIS cleavage, the distinction between these syndromes is likely to lie in tissue specific differences in substrate cleavage and collagen post-translational modification and crosslinking" sounds vague and seems not to be supported by the data. From the results, one might rather suspect, that OASIS and, eventually, ATF6 are not the crucial substrates affected by the OI-specific MBTPS2 mutations.

We thank the Reviewer for these constructive observations, which have informed our revised Discussion.

Major suggestions:

1) The last part of the discussion, dealing with the mechanism leading to OI versus phenotypes of the IFAP complex, might be clarified to differentiate between conclusions supported by the results and speculations calling for further studies.

The new urinary crosslink data does bring the two X-OI Pedigrees in line with each other for pathomechanism, but the pathways that distinguish X-OI and IFAP remain to be elucidated. We have revised our Discussion along these lines on pgs. 14-15, citing both the possibility of tissue-specific differences in pathways and of novel RIP substrates critical to bone development

2) In the Methods chapter, the authors describe Luciferase experiments with p5xATF6 or pSRE reporters. Results with pSRE are not shown, but might be interesting in view of the question if the S2P substrates ATF6 and OASIS employed in this research are the critical ones leading to X-linked OI.

Please see response to Reviewer 1, Major Point #3.

3) Page 22: Legend to Fig. 6c: References for the results, on which this schematic is based, should be given.

The References have been added, refs. 3,5,13, and new reference #21.

4) Supp. Fig. 1 and Fig. 5b: Individuals IV5 and IV21 are described as possible carriers (Supplemental Fig. 5b). IV5 does not have the haplotype of mutated chromosome. IV21 is shown in Supp.Fig. 1 as carrier, based on the X-inactivation result only.

For Supplemental Figure 1, the status of female members is based on mutation analysis by RFLP. IV5 has only the undigested 680 bp band, indicating that her genotype is Wt/Wt, while IV21 has the undigested 680 bp band and the digested 375 and 305 bp bands, indicating that her genotype is Wt/Mt. In the original Supplemental Figure 5b, the status of female members was based on pedigree only.

To eliminate this confusion, Supplemental Figures 5b and 1 are now in agreement, with IV5 listed as normal and IV21 listed as obligate carrier. We have added a sentence to the Legend of Figure 5b stating that the status of the females is based on mutation analysis.

Minor:

Page 3: Reduced collagen crosslinks presumptively undermines bone strength....
→..undermine...

Done

Page 8: ...predict both MBTPS2 variants p.N459S and p.L505F as pathogenic... (Do the prediction programs determine the mutations as pathogenic or possibly pathogenic?)

Page 12: dito!is predicted to be pathological by multiple mutation analysis programs...
→...possibly pathogenic...

We have changed the sentence to read "Multiple programs (Polyphen2, SIFT, Provean, MutationTaster) predict both MBTPS2 (S2P) variants p.N459S and p.L505F as pathogenic since they are likely to interfere with function"

Page 17: Reference Naiki et al. is incomplete.

Now completed.

Page 22: Legend to Fig. 6c: References ... on which this schematic is based, should be given.

Done. See also response to Reviewer #1, Minor Suggestion #3.

Supp. Page 9: ...according to the manual instructions...→ ...to the suppliers instructions...

Done. On page 8 in revised Supplement.

Supp page 17: ... The MBTPS2 c.1376A>G (p.N459S) mutation segregated with phenotype in all members. →...all affected members?...

Done

Supp. Page 21: ...(b) Levels of X-inactivation of female members were shown.... →..are shown...

Done

Supp. Pages 22 and 24: Supp Figs 6 and 8. It may be helpful to explain "NL" in the legend (Normal).

Done

Supp. Page 33: Supp. Table 6: According to "Methods" luciferase assays have been performed for p5xATF6 or pSRE reporters. What is shown here? A bit more of legend would be helpful.

If it shows only the result of an assay with p5xATF6, where is the result of pSRE shown?

In the Revision, Supplemental Table 7 has the ATF6 results, and Supplemental Table 8 has the SRE results.

Supp.Pag3 34: Ref.2 is incomplete.

Done

We thank the reviewers for their time and constructive critique, which has been helpful in improving our manuscript.

Reviewer Clarification:

There were 5 carriers in the Thai pedigree. We have corrected the number on page 7, to match Supp. Fig. 5.